# Bend or PIN: Studying Bend Password Authentication with People with Vision Impairment

Daniella Briotto Faustino*
Carleton University

Sara Nabil†
Carleton University

Audrey Girouard‡
Carleton University

## ABSTRACT

People living with vision impairment can be vulnerable to attackers when entering passwords on their smartphones, as their technology is more 'observable'. While researchers have proposed tangible interactions such as bend input as an alternative authentication method, limited work have evaluated this method with people with vision impairment. This paper extends previous work by presenting our user study of bend passwords with 16 participants who live with varying levels of vision impairment or blindness. Each participant created their own passwords using both PIN codes and BendyPass, a combination of bend gestures performed on a flexible device. We explored whether BendyPass does indeed offer greater opportunity over PINs and evaluated the usability of both. Our findings show bend passwords have learnability and memorability potential as a tactile authentication method for people with vision impairment, and could be faster to enter than PINs. However, BendyPass still has limitations relating to security and usability.

**Index Terms:** Human-centered computing—Human computer interaction (HCI)—; Human-centered computing—Haptic devices—; Human-centered computing—User studies—;

## 1 INTRODUCTION

While accessibility features like screen magnifiers and screen readers make devices such as flat touchscreen smartphones usable for people with vision impairment, many challenges remain. Typing on smartphones, for example, is complex for users who are blind or have low vision [4, 30], often requiring them to use external physical keyboards [6, 47]. Also, screen readers read everything aloud to users, jeopardizing their privacy and requiring them to use of earphones in public spaces [3, 7, 47]. Additionally, accessibility features have a drawback of making vision-impaired users vulnerable to shoulder surfing and aural eavesdropping when entering PINs [23]. Shoulder surfing can result from the use of screen magnifiers zoom in the keyboard, making password entries more visible, while aural eavesdropping is possible because screen readers read everything typed aloud, even password entries. Therefore, almost 70% of the vision-impaired are concerned with typing passwords in public spaces and prefer biometric user authentication, such as fingerprints [15]. However, biometrics may not work when there is an environmental change (e.g. moist hands), and thus act as a re-authenticator for other authentication methods, such as PINs or patterns [5]. The problem is twofold: patterns are not accessible for people with vision impairment [8, 28] and PINs are regarded by them as insecure to unlock mobile devices [8, 15], easy to guess, inaccessible or inconvenient [7, 15, 18].

To give vision-impaired users a more accessible and secure alternative for patterns and PINs, previous work has suggested the

---

*e-mail: daniella.briottofaustino@carleton.ca
†e-mail: sara.nabil@carleton.ca
‡e-mail: audrey.girouard@carleton.ca

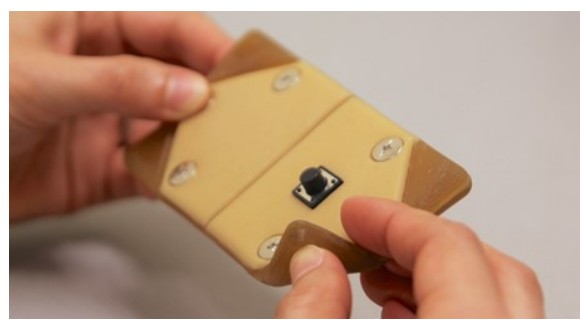

Figure 1: BendyPass prototype, where users can enter bend and fold gestures to compose a bend password.

use of deformable user interactions, which supports tactile interaction [36, 42, 43]. More specifically, they proposed the use of bend passwords [33], a user authentication method where a sequence of bend gestures works as a password. However, no previous studies were carried out to investigate the usability of bend passwords for the vision-impaired.

In this paper, we present our exploration-led research of bend passwords. For our study, we developed BendyPass, a flexible device to capture bend passwords, that was proposed by in a previous demonstration [14] and extended its capabilities into an interactive device (Figure 1). We developed it through an iterative process with deformable interfaces' researchers and vision-impairment experts. Then, we conducted a within-subject study with vision-impaired participants to compare bend passwords and PINs to answer these research questions:

- Q1. Would deformable interaction outperform touch interaction for people with vision impairment?

- Q2. What can we learn —for design— about the *ease of use*, *memorability* and *learnability* of bend passwords versus PINs for people with vision impairment?

Our paper builds on two main publications [14, 33] in three key aspects. We created a refined prototype (inspired from a demonstration [14]) that is more efficient and robust with additional features. We targeted a specific user population, people with vision impairment, that no previous work has actually run studies with/for. We evaluated both bend and PIN authentication concurrently to quantitatively compare them and explore the usability of each in a rigorous and thorough user study, breaking new grounds.

As such, our study is the first to explore the potential and limitations of bend passwords with people with vision impairments. Through the analysis of the data collected from 16 blind and low vision participants, our three key contributions are: (1) Refining the design and fabrication of deformable prototypes for password input; (2) Presenting insights on the usability of bend passwords compared to PINs; (3) Exploring the design opportunities and challenges of bend passwords.

## 2 RELATED WORK

Researchers have explored the use of technology for assisting people with vision impairment [4, 17, 23]. While mostly relied on touch interaction on common smartphones, some researchers explored the opportunity of authenticating through physical deformation rather than touch-screens, for people living with blindness or vision impairment. In this section, we review the use of technology, authentication methods and deformable interfaces with respect to such special needs.

### 2.1 Technology and Vision Impairment

Smartphones have become widely adopted not only by sighted individuals, but also by individuals with vision impairments, thanks to the rise of accessibility features and assistive applications in mainstream devices [17, 23]. The most common accessibility features are screen readers and screen magnifiers. With the increased smartphone adoption, the quality of life among the vision-impaired has improved [4], because they use their smartphones to execute tasks previously achievable only by using multiple assistive technology devices. Smartphones work as assistive tools aggregators, giving users access to apps to identify bills [11], street names [12], colours [16], objects and faces [35], read printed text [1].

Nonetheless, using smartphones also has its own challenges. Entering data on smartphones can be difficult for the vision-impaired [4, 30], who might need to use external physical keyboards [6, 47]. Also, screen readers put users at risk of others listening to their private information, requiring the use of earphones in public spaces [3, 7, 47]. Braille is usually considered a solution, because eventual attackers probably do not know how it works. However, few vision-impaired know Braille. Among vision-impaired Americans and Canadians, fewer than 10% read Braille [37, 38] and among the British, less than 1% of the people with vision impairment can read Braille [41].

### 2.2 User Authentication Methods

User authentication methods are the way users can prove their identity, have access to their devices and accounts [33], and protect their personal information [27]. The user authentication methods can be categorized as: knowledge-based (something the user knows, such as PINs passwords), token-based (something the user has, such as key fobs), or biometric-based (something the user is, such as fingerprints) [27]. Generally, alphanumeric passwords are the standard for user authentication on websites [13]. On the other hand, PINs are the standard for unlocking smartphones [45]. For sighted Americans, for example, PIN is the most commonly used method to unlock smartphones [40].

A 2018 online survey with 325 vision-impaired people found 75% the smartphone users had an authentication method to unlock their devices [15], a higher percentage than the ones found in research from 2012 [7] and 2015 [18], where 0% and 33% used an authentication method, respectively. The 2018 survey also found their preferred user authentication method on smartphones is fingerprint, which they also consider the most secure authentication method [15]. On the other hand, participants considered PINs the least secure authentication method, and only 16% use it as their main method [15]. The perceived security of PINs is impacted by the fact that typing PINs when using embedded screen readers makes people with vision impairment more susceptible to others aural-eavesdropping [23]. Similarly, using screen magnifiers while entering passwords increase the susceptibility for visual eavesdropping or shoulder surfing [23]. However, most vision-impaired seem unaware that even if a smartphone has a biometric method set up, a knowledge-based authentication is still their main barrier against unauthorized access to smartphones [15], as biometrics act as mere re-authenticators for knowledge-based methods [5].

Vision-impaired users do not want to deal with the complexity of user authentication methods [15], but with the large volume of personal data generally stored in smartphones [27], it is essential to protect users' privacy [23]. Researchers have explored user authentication alternatives for the vision-impaired to reduce observer threats, including password management system [10], accessible pattern for touch-screens [8], and novel user authentication methods based on the user's gait [23] and multi-finger taps on the touch-screen [7]. However, no previous study has rigorously explored the use of deformation gestures as a method for user authentication with people with vision impairment yet.

### 2.3 Deformable Bendable Interfaces

Deformable devices allow users to physically manipulate their shapes as a form of input, by bending, twisting or deforming them [2, 20]. Such interfaces are often referred to as Deformable User Interfaces (DUIs) [24] - that is, the "physical manual deformation of a display to form a curvature for the purpose of triggering a software action", including a bend gesture [26]. One of the first interfaces designed to explore physical input interaction through bending was ShapeTape [9]. Similarly, Gummi [42] was made to afford bending its physical form using flexible electronic components including sensors that are able to measure deformation. Companies (such as Samsung, LG and Huawei) are currently developing foldable flexible devices, and we expect deformable devices to include bend gestures as input in the near future.

Considering how blind users solely rely on non-visual feedback (e.g., tactile cues and audio), and that deformation is a tactile form of input, Ernst et al. [20] proposed deformation could be beneficial for the blind. They developed a deformable device that accepted bend gestures to control a smartphone screen reader. In their preliminary evaluation with vision-impaired participants, Ernst and Girouard [19] found bend gestures might be "easier out of the box than touch [interactions]" and could improve the accessibility of smartphones.

Another possible application for bend gestures is bend passwords, first proposed by Maqsood et al. [33] as a sequence of bend gestures to authenticate the user. In a user study with sighted participants comparing bend passwords to PINs, researchers found bend passwords easy to memorize as PINs, but might allow users to rely on their muscle memory to recall their passwords [33]. Additionally, bend passwords might be harder to observe than PINs, being potentially safer against shoulder surfing attacks [32]. Our work expands and evaluate a prototype published as a demonstration [14]. Due to the tactile nature of this method and its promising opportunity, and as flexible smartphones may become available soon, we want to explore the usability of bend passwords for people who are vision-impaired or blind.

## 3 PROTOTYPE

This section describes the design process resulting in our final prototype that we later evaluate with people with vision impairments. The related previous work, especially [14, 33] were the launching points for this paper that helped: 1) conceptualize and demonstrate the technical issues of creating an initial prototype [14]; and 2) test this interaction method with a generalized population (i.e. sighted people) [33].

### 3.1 Design Process

We started by consulting a blind expert who teaches technology to people with vision impairment in a local organization of the blind. The expert shared her concerns about vision-impaired users deciding not to use authentication methods on their smartphones due to their perceived inaccessibility or complexity, and confirmed that bend passwords could have potential for the vision-impaired. Then, we developed an initial version of the prototype and presented it to a group of 10 vision impaired, describing bend passwords as an

alternative for authenticating. Most of them indicated interest in a tactile user authentication method.

Our process consisted of developing concurrent alternative prototypes based on previous research and on feedback received from deformable interface researchers of the demonstration prototype, then presenting them to the two blind experts to participate in our iterative design process. We consulted them to define the best size, material stiffness, groove design, sensor placement and set of gestures, to create a prototype that would be well suited for such special needs. We iterated through a dozen preliminary prototypes and a total of four meetings with the blind experts before choosing our final design.

### 3.2 Prior Work Inspiration

Previous research showed that users prefer a deformable device the size of a smartphone rather than a tablet, to minimize the user's level of fatigue and comfort [29] and minimize the need to reposition their hands to perform bend gestures [19, 31]. Users perform most deformations while holding a rectangular device horizontally [28] and they select simple bend gestures that are less physically demanding [26, 28]. Additionally, forcing the user to change their grip to perform gestures not only causes discomfort [22, 28], but also slows task completion [2, 19], raises the likelihood of false activation [22], and increases the risk of dropping the device [22]. Hence, we decided to design a rectangular device similar in size to a medium smartphone and considering the importance of allowing users to access all corners without re-gripping, we designed our interface for landscape use.

Regarding device deformability, previous research showed that higher stiffness requires more physical effort [25], which negatively influences users' preference and performance when bending [24]. Finally, researchers recommend that bendable interfaces could help users identify deformable areas by having grooves on the bendable points [19, 22] and by providing haptic feedback [43]. Thus, we opted for using malleable silicone for our device's bendable areas, adding grooves and haptic feedback.

### 3.3 Final Prototype Design

This prototype's purpose is to put this opportunity in the hands of users and explore its potential in comparison to touch-screens. Our final prototype, BendyPass (Figure 1) is approximately the size of an iPod Touch (11.5 × 6 × 1 cm), made of silicone and has a single push-button that allows the user to either confirm the password or delete the previous gesture entered. The button also indicates the device's orientation, both helping users to recognize which side should be facing them, and which side should be left or right.

Inspired by a recent demo [14], our prototype is composed of two silicone layers that enclose all electronic components. We 3D printed a mold including a vertical groove in its centre, four corner grooves that create triangular areas around each of its corners, and a lowered part to insert its push-button (Figure 2, top). The grooves extend from side to side to facilitate bending gestures and they are also angled to avoid users' fingers to be pinched when bending the device. To fabricate our prototype, we used two different types of silicone, making the bendable areas more flexible than the central area (Alumilite A30 and A80, respectively). This emphasizes the bendable areas while protecting the components in the center.

The electronic components included 5 1" Flexpoint bidirectional sensors to recognize bend gestures and a vibration motor to provide haptic feedback when an action is recognized. BendyPass components are positioned as shown in Figure 2 (bottom): the vibration motor is on the left side and the flex sensors are in the centre, and in the four corners. The components are connected to an Arduino Leonardo microcontroller connected to a MacBook Pro laptop. Gestures applied to our prototype become letters on the computer, while long button presses (¿ 1s) activate the Enter key and short button

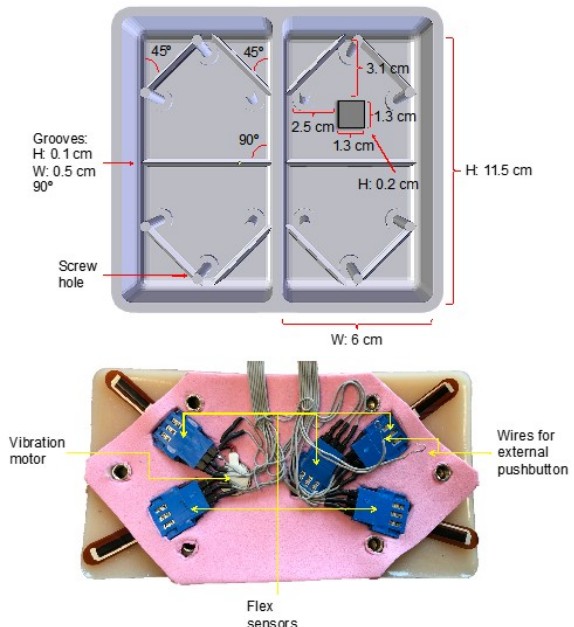

Figure 2: BendyPass 3D mold design (top view) and its internal components housed in a thin foam layer (pink).

presses activate the Backspace key. The letter mapping is invisible to the user, who only needs to perform bend gestures and operate the button.

### 3.4 Bend Passwords on BendyPass

BendyPass recognizes 10 simple bend gestures (Figure 3), including bending each corner upwards or downwards (8 gestures) and folding it in half upwards or downwards (2 gestures). Our prototype was programmed to recognize fewer gestures than Maqsood et al.'s [33], because our two blind experts considered excessively complex the double gestures (gestures performed in two corners at the same time). With 10 possible gestures, BendyPass has the same number of possible combinations of a PIN, as a six-gesture bend password has the same strength against brute force guessing as a 6-digit PIN.

Aside from the haptic feedback, BendyPass also provides optional audio feedback verbalizing the name of the gesture entered. Prior work indicated that for an effective communication of gestures, it is necessary to provide information about location and direction [44]. Thus, we named our gestures using both (e.g. "top left corner up"). The exceptions are folding gestures, which showed to be confusing in preliminary analysis using location. For example, a "centre up" could trick users into moving the sides up, while the centre would move down. Thus, we opted to use the name of the gesture—fold—in addition to the direction for these.

## 4 USER STUDY

### 4.1 Apparatus

We designed a user study to compare two knowledge-based user authentication methods: bend passwords and PINs. Thus, in addition to our prototype BendyPass, we used an iPhone 6S for PIN entry, because most people with vision impairment use iPhones [15] due to its native screen magnifier and screen reader functionalities. We selected a keypad from a remote keyboard app [52] to simulate a PIN entry screen while transmitting the keys typed to a computer, where we could save them. We chose it because it worked relatively well with the screen reader VoiceOver and the Standard typing style.

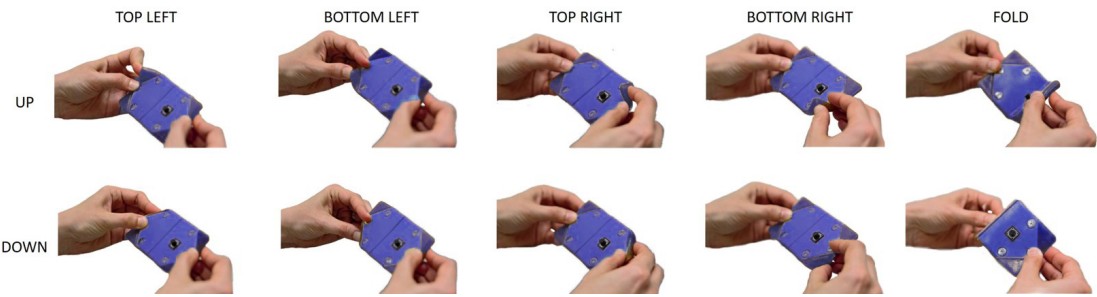

TOP LEFT   BOTTOM LEFT   TOP RIGHT   BOTTOM RIGHT   FOLD

UP

DOWN

Figure 3: Set of 10 bend gestures available on BendyPass.

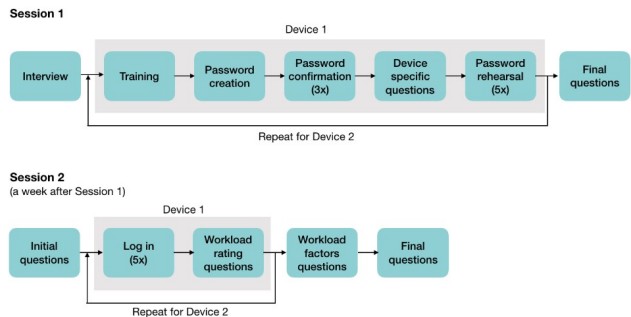

Figure 4: The structure of our user study.

In this accessible typing mode, users explore the screen by either swiping or single-tapping it and they trigger a key by either lifting their finger and double tapping the screen or keeping their finger on the screen and tapping it with another finger.

We also developed a PHP website to receive and verify passwords both from BendyPass and the smartphone. The website was connected to a mySQL database and hosted using XAMPP. Our database saves participants IDs, password entries, and the time the input started and ended. Our website provides audio cues (such as "Create your password using the gestures learned" or "Wrong password, please try again") to help users navigate the password creation process. It also provides optional audio feedback for bend gestures or button presses. For the audio, we recorded the screen reader VoiceOver reading all messages on a MacBook Pro, using the default speed of 45. We used JavaScript to assign actions to their respective audio snippets.

## 4.2 Methodology

We structured our user study to be composed of two 60-minute sessions (Figure 4), following the main tasks proposed by relevant literature [33]. The first session focused on password learnability, while the second session, about a week after the first, focused on password memorability.

We started the first session by interviewing participants, asking them whether they had already tried a flexible device, followed by questions on their demographics and user authentication perception and use. Then, we asked participants to create both: 1) a bend password on BendyPass and 2) a new PIN on the smartphone with at least 6 digits/gestures. After creating the passwords, participants confirmed them 3 times, before rehearsing them by completing 5 successful logins. Whenever participants expressed uncertainty about their passwords, while confirming or rehearsing them, we asked if they wanted to create a new one and allowed them to do so. After confirmation, participants had a pause to answer questions about the

easiness to create and remember their password, and the perceived overall security, specifically against shoulder surfing. Participants could create a new password if forgotten. Participants who chose to create a new password during rehearsal had to immediately confirm and rehearse them. The session ended with questions allowing participants to reflect on their experience, the likelihood of using bend passwords and further insights about what worked well and what did not.

After a week, we started the second session by asking participants questions related to their easiness to remember their passwords, whether they used any strategy to memorize their password during the week, which accessibility features and typing styles they use in their own smartphone. Then, we gave participants as many attempts as they needed to complete 5 successful logins using each of the two passwords created in the first session. Finally, we finished the session by discussing with them their final thoughts and reflection regarding their likelihood to use bend passwords over PINs (in general and in flexible devices), their overall experience using BendyPass, potential user groups for bend passwords, and any proposed enhancements.

We presented the two devices to participants in counterbalanced order, among participants and between sessions with the same participant. In both sessions, after interacting with each device, participants answered device-specific questions. Besides learnability and memorability, we also evaluated the other quality components of usability [39], by measuring efficiency and satisfaction in session 2, and number of errors in both sessions.

In both sessions, participants verbally answered our questions about each device, regarding the easiness to create passwords, the perceived security of both password schemes and their opinions about bend passwords (Figure 5). All questions were 10-point Likert scales, where 1 was the least favourable.

## 4.3 Data Analysis

We transcribed participants' comments and answers to open-ended questions using Inqscribe (inqscribe.com). We performed the qualitative analysis on open-ended answers and quantitative analysis on both multiple-choice and coded answers using R Studio (rstudio.com). Quantitative analysis included the analysis of 732 log records (participant $\times$ step $\times$ trial) using Wilcoxon Signed-Rank (Z) for numerical data, and chi-square tests ($\varkappa^2$) of independence between variables for categorical data, but we focus on reporting significant results. Time measured included time thinking about the passwords and time entering them. We used Grounded Theory [18] to code answers of interview questions. Whenever necessary, we coded answers in more than one theme, but we did not code unclear answers.

## 4.4 Participants

We recruited participants by snowballing, mainly through a local council of the blind and Facebook groups. Our recruitment criteria

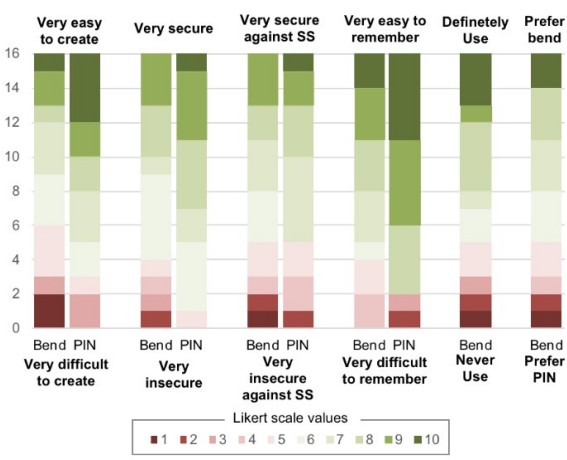

Figure 5: Distribution of Likert scale responses; the first three pairs are from session 1, the last three are from session 2. SS stands for Shoulder Surfing.

included participants who were at least 18 years old and were either blind or had low vision. We held sessions either at a lab or at an office at the local council of the blind. Among our 16 participants, 10 declared they were blind, 5 declared they had low vision, and one expressed having another condition. For our analysis, we grouped the latter with the blind because he could not see the smartphone screen. This resulted in 11 blind participants (68.7%) and 5 with low vision (31.2%), a distribution similar to previous work [15]. Many participants self-declared as males (N=10, 62.5%). Ages of participants ranged from 22 to 76 years-old (M=54.31, SD=15.38). Three participants also had another impairment, related to hearing loss (N=2), attention (N=1) or psycho-motor system (N=1), according to the World Health Organization classification [46].

Almost all participants said they use assistive apps on their smartphones (N=15, 93.8%), only 5 participants said they use a Braille display, a smaller proportion than in relevant work [15] (31.3% vs 42.5%). Three participants had experience in studies on deformable flexible devices, though never for user authentication. Most answers to the interview matched results from the survey in prior work [15], suggesting our participants represent well the group of people with vision impairment who have access to the internet and mobile devices. All blind and most low-vision participants interacted with the smartphone with a screen reader. Only one participant used screen magnifier. All participants returned for session 2 about a week later (M=7.28 days, Md=7, SD=0.87).

## 5 FINDINGS

We discuss each of our main findings, introducing potential and implications for design learnt from studying bend passwords in-use with users with vision impairment. Our reported findings combine the results from observation of usability, analysing log files of passwords, and questionnaire results.

### 5.1 Learnability

#### 5.1.1 Ease of Creating

Before creating their passwords, participants trained for a longer period to use bend gestures (M=165s, Md=143.5s, SD=63.94s) than they trained to use the keypad app (M=90s, Md=91.5s, SD=30.92s), (Z= -2.97, p ¡ .005). After training, participants took slightly longer to create their first bend password (M=59.6s) than their first PIN (M=48.8s), but the difference was not statistically significant (n.s.),

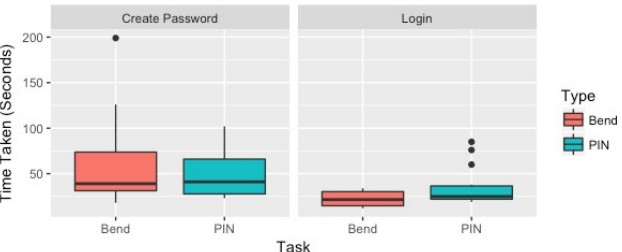

Figure 6: Creation and Login time in the first trial, in seconds. Differences are not statistically significant.

as shown in Figure 6. Moreover, participants' perceived easiness to create bend passwords was not statistically significantly different than creating PINs (Table 1). We also found no significant differences between participants who are blind and who had low vision regarding learnability (n.s.).

#### 5.1.2 Password Creation Strategies

We observed how participants created passwords and asked which strategies they used to create them (Table 2). More than half of our participants used some sort of pattern (N=9), such as mirroring gestures from one side of the device to the other (N=5). Our results are similar to those from previous work [33] with sighted participants, where patterns were the main strategy used to create passwords (44%). Although almost half of our participants said they associated their PINs to series of numbers they were familiar with (N=7), no participant reported using association as a strategy to create their bend passwords.

### 5.2 Memorability

#### 5.2.1 Re-thinking Passwords

Although participants simply created their first bend password, most did not remember it. 11 participants had to re-create their bend passwords, while only 2 had to re-create their PINs, resulting in a significant difference between the number of trials to create memorable bend passwords (M=1.94, Md=2, SD=1) and PINs (M=1.12, Md=1, SD=0.34) (Z= -2.36, p ¡ .01). From the 11 participants who forgot their initial bend password, 9 forgot it at the confirmation step (81.8%), and 2 forgot it at the rehearsal step (18.2%). We attribute the initial difficulty to memorize bend passwords to the lack of muscle memory for bend passwords, which required participants to create new memorization strategies. As P5 said, *"It's fun but you have to suspend anything you know about passwords. You have to think in a new way."* As participants had to confirm passwords 3 times after creating them, P13 said that having to go through three confirmations probably makes passwords easier for people to remember, *"engraving them into memory"*, and suggested that this should be required in real-life.

#### 5.2.2 Ease of Recall

In session 2, although 81.3% (N=13) participants remembered their bend passwords, 1 participant was not able to enter it due to prototype errors. Thus, participants' login success rate was 75% (N=12) for bend passwords and 93.8% for PINs (N=15), but a McNemar test with the continuity correction found no significant difference between them. Most participants successfully entered their bend passwords and PINs in the first trial (n.s.). Although the memorability of PINs was slightly better, bend passwords had a similar memorability (n.s.), even though it was a novel method for participants. Additionally, while in session 1 participants rated bend passwords significantly harder to remember than PINs (Table 1), their ratings

| Question | Session | Bend Md (SD) | PIN Md (SD) | Statistics |
|---|---|---|---|---|
| Ease of password creation | 1 | 6.0 (2.62) | 7.5 (2.33) | $Z= -0.95$, p = .17 |
| Ease of remembering | 1 | **5.5 (2.66)** | **8.0 (1.68)** | **$Z= -1.88$, p = .03** |
|  | 2 | 7.5 (1.98) | 9.0 (2.38) | $Z= -0.97$, p = .17 |
| Confidence in remembering | 1 | 7.0 (2.47) | 8.0 (1.89) | $Z= -1.56$, p = .06 |
|  | 2 | 8.0 (1.89) | 9.5 (2.68) | $Z= -0.86$, p = .20 |
| Perceived overall security | 1 | 6.0 (2.13) | 8.0 (1.46) | $Z= -1.40$, p = .08 |
| Security against shoulder surfing | 1 | 6.5 (2.37) | 7.0 (2.15) | $Z= 0.68$, p = .75 |
| Likelihood to use if available | 1 | 7.5 (2.83) | - | - |
|  | 2 | 6.5 (2.69) | - | - |
| Likelihood to use in flex. devices | 1 | 6.5 (2.49) | - | - |
|  | 2 | 5.5 (2.53) | - | - |

Table 1: User Questionnaire responses to 10-point Likert scale questions, where 1 represents strongly disagree. Significant difference marked in bold.

| Strategies to create passwords | Bend passwords | PINs) |
|---|---|---|
| Pattern | 5 (9) | 4 (9) |
| Simple combination | 5 (3) | 0 (2) |
| Repetition | 0 (3) | 2 (6) |
| Association | 1 | 7 |

Table 2: Strategies participants reported using to create memorable passwords. Numbers in parentheses express the number of times researchers observed the use of each strategy.

| Type | Mean Length (SD) | Mean Unique Entries (SD) |
|---|---|---|
| Bend Password | 6.44 (0.81) | 5.81 (1.05) |
| PIN | 6.19 (0.54) | 4.69 (1.14) |

Table 3: Unique entries are the number of unique digits (PIN) and gestures (bend password) in a password.

in session 2 for the same questions about bend passwords and PINs were not significantly different (n.s.).

### 5.2.3 Confidence to Remember

Participants' confidence to remember their passwords was affected by the number of errors they made in the rehearsal step. Those who had fewer incorrect trials rated their confidence significantly higher both for bend passwords ($x^2$ (24, N=16) = 36.98, p = .04) and for PINs ($x^2$ (10, N=16) = 23.43, p = .009). Also, participants who were more confident in remembering their passwords before login in session 2 were significantly more likely to remember their bend passwords ($x^2$ (6, N=16) = 85, p = .01) and their PINs ($x^2$ (5, N=16) = 85, p = .007).

### 5.2.4 Password Memorization Strategies

We asked participants in session 2 whether they used any strategies to remember their bend passwords. Four (25%) said they thought about their passwords throughout the week for a total of 7 (43.8%) who thought about them at least once. Also, while 7 (43.8%) said their methods to create their bend passwords were the main strategy to memorize them, 2 (12.5%) said they did not use any strategy. Both of them forgot their bend passwords in session 2 and a participant who forgot her PIN also did not use a strategy to remember it. Association, a common strategy used to memorize PINs, was not used by participants to memorize bend passwords, potentially because of their three-dimensionality. However, results from session 2 indicate that, regardless the strategy used to create passwords, maintain them in one's memory depend on good memorization strategies, which include at least repeating in one's head how the password was.

### 5.2.5 Rate of Errors

Our analysis of memorability considered the number of correct logins in session 2. Thus, for analysing the number of errors, we considered the errors that were not followed by a new password creation, excluding only those caused by a prototype error. Both bend passwords and PINs had the same number of incorrect entries (N=7). Similarly, the number of participants that made incorrect entries was the same: 4 for each password type (only 1 participant had an error with both bend passwords and PINs). Thus, there was no significant difference in number of errors (n.s.).

## 5.3 Ease of Use

Participants rated the likelihood of using bend passwords for different users Most participants considered BendyPass easy to use (N=10) and liked its haptic and audio feedback (N=9). We also analysed the bends used to compose each password.

### 5.3.1 Potential Users

When asked who might like to use bend passwords, 12 (75%) participants said vision-impaired people in general. P5 said, *"Certainly blind and low vision, or people with learning disabilities that make them have issues with numbers, and seniors or people with learning issues"*. This supports the idea that physical deformable interaction in general, and bendable interfaces in particular, are perceived to offer great potential to people with low vision, not only by researchers [15, 19, 33], but also by users themselves.

### 5.3.2 Bend Passwords Used

We analyzed the password characteristics and the composition of all passwords created by our participants (Table 3). Both bend passwords and PINs ranged from 6 to 8 gestures/digits, but most were equal to the minimum length of 6 required from participants. Password length was not statistically significant. In contrast to prior work [33], our participants used more unique gestures than unique digits ($Z= -1.95$, p = .03). Every bend gesture was used at least once by at least one participant to compose a bend password. However, some gestures were more frequently used than others. The top three most frequently used gestures were: top right corner up (17%), top left corner up (14%), and bottom left corner up (12%), exactly the same top three single gestures for sighted participants [33]. The least used gestures were top right corner down (6%), fold down (7%) and fold up (8%). Participants tended to prefer upwards gestures (60.2%) than downwards gestures (39.8%), confirming previous findings [26, 31, 33], even though the difference was not significant.

## 5.4 Satisfaction

To evaluate the satisfaction from a user perspective, we studied the time needed to login, measuring efficiency, asked participants about

the overall experience and perceived security Finally, we also report below on their perceived drawbacks and limitations.

### 5.4.1 Overall Experience

After the final session, we asked participants to tell us how they would describe their experience using bend passwords to a friend. Nine participants expressed positive experiences using BendyPass while only 3 described it negatively. For example, P4 said it was *"fun, interesting, challenging, intriguing"*, while P8 said, *"it was easy, [there is] a little of learning curve to know how to do the bends right, but once you got that it's easy to use, even easier than swiping the touch screen to find numbers"*. P7, on the other hand, said, *"if errors were removed, it would be OK. Primary reason for negative comments are the errors and the fact that the surface should be [...] more responsive."*. The errors mentioned by the participant involved non recognition of gestures performed (N=5) or the recognition of opposite directional gestures (N=1). The latter was caused by the sensors used in BendyPass, which react not only to deformation but also to pressure, which can occur when participants grasp the prototype strongly.

### 5.4.2 Efficiency of Login

Following the methodology used in relevant literature [33], we compared participants' fastest confirmation and rehearsal times to evaluate whether their performance improved with practice. Participants took significantly less time to rehearse their PINs than to confirm them (Z= -2.97, p = .001), but were not significantly faster with bend passwords (n.s.), indicating their efficiency achieved optimal numbers from the start. Participants took slightly less time to complete a first login with their bend passwords (M=22.4s, Md=21.5s, SD=8.48s) than with their PINs (M=35s, Md=25s, SD=21.25s) (n.s.), as shown in Figure 6. We selected the fastest login time from each participant who logged in successfully. As observed in the previous steps, participants took significantly less time in their fastest login with bend passwords (M=13s, Md=12s, SD=3.1s) than with PINs (M=18.27s, Md=16s, SD=7.71s) (Z= -2.20, p = .01).

### 5.4.3 Perceived Security

We found no significant difference between the perceived security of bend passwords and PINs. Interestingly, when we asked participants to justify their ratings for the security of bend passwords and PINs against shoulder surfing attacks, 7 participants said bend passwords are easy to see by others, while 5 participants said PINs are easy to see. This was also one of the most common reasons participants gave for their ratings of the overall security of bend passwords (N=5), although another 4 participants said bend passwords were difficult to see. Thus, there was no consensus amongst participants on whether bend passwords are easy or hard to observe.

### 5.4.4 Drawbacks of Using BendyPass

We asked participants to point out characteristics that worked well with bend passwords, as well as aspects that should be improved. The most common disadvantage participants mentioned was having to carry an additional device (N=6). For example, P7 said *"I don't like carrying extra things, I barely remember my charging cable"*. On the other hand, most suggested reducing the protuberance of the button (N=11) and improving the accuracy of the bend password recognition (N=10), especially for folding gestures (N=9). Although more than half of the participants liked the audio feedback provided by BendyPass, at least 3 inclined towards deactivating this option. Interestingly, 7 participants liked the form factor of BendyPass while 8 suggested the reduction of its size, even saying it would be nice to have it as a key chain.

### 5.5 Limitations

Similar to prior work [15], we recruited more blind than low-vision participants. This might be a result of a higher interest of the blind community in novel assistive technologies but might also be an indicative to the difficulty in classifying some people as blind or low vision. For example, one of our participants self-declared as blind, but said he uses inverted colours on his smartphone to better see the screen.

Although we tested our prototypes and had 3 pilot sessions, we faced prototype issues during the study sessions, where BendyPass was not fully accurate on recognizing bend gestures, and the smartphone app did not work with all typing styles. Although we acknowledge both the app keypad and the typing method in our study are not the same as most participants use in their own smartphones, we argue the learning process new users go through when using a new device or software was simulated well.

## 6 DISCUSSION

We presented the results of a user study on bend passwords compared to PINs with 16 participants who were blind or had low vision. We found that bend passwords were as easy to create as PINs, and participants assessed them as easy to remember as PINs. Participants reported being more likely to use bend passwords on a flexible device but would also be willing to use them in a separate device for gaining access to password-secured systems and spaces. This is because bend passwords take users with low vision less time to enter than PIN passwords.

### 6.1 Learnability of Bend Passwords

Explaining to participants how to use BendyPass took around 2 minutes, confirming previous findings [42] that users are able to quickly understand how to interact with deformable devices. Users may need more training with BendyPass than with the smartphone, and slightly more time to create bend passwords than PINs. We attribute this to the novelty of the paradigm as opposed to the commonly used touch-based PIN passwords. Still, our results show that the time needed to create a password using bend or PIN are not statistically significant.

### 6.2 Memorability of Bend Passwords

Participants forgot their bend passwords significantly more often than their PINs. This can relate to their unfamiliarity memorizing gestures compared to memorizing sequences of numbers. The initial difficulty to memorize bend passwords is supported by participants' ratings during the first session on the easiness to remember bend passwords, significantly lower than PINs. However, the results of the same question in session 2 were not statistically significant. This is also supported by their success rates and the number of attempts to complete a first successful login. Still, the lack of muscle memory for bend passwords that supports the association memorability strategy impacts the performance of bend authorization compared to PINs.

Perhaps future work should explore muscle memory associations such as typing on a keyboard or playing a musical instrument. Other deformable authentication methods can be also used to support people with vision impairments using mental association for better memorability. Examples of such interactions could mix both deformation with alpha-numeric associations, such as enabling only a small number of bend gestures in a Morse code style sequence, stroking characters on a texture-changing edge, or twisting a shape-memory strip a number of times back and forth.

### 6.3 Easiness to Use Bend Passwords

Users rated bend passwords as easy to use and appropriate for people with vision impairments. Bend passwords were faster to login than PINs for people with low vision. This is mainly because our participants used smartphones with accessibility features, which slowed

them down. In fact, research acknowledges that being slow to enter is actually one of the great limitations users dislike about PINs [15].

### 6.4 Usability for Blind vs Low Vision

We did not find significant differences between the time participants who were blind and participants who had low vision took to train how to use bend passwords or to create a bend password. This suggests that people with low vision can benefit as much as the blind from a tactile physical password.

### 6.5 Overall Usability of Bend Passwords

Due to the familiarity with PINs—as expected—, the time needed for learning and training how to use bend passwords can take longer. PINs are often created from numeric patterns that users' remember, such as dates and phone numbers that are already saved in our memory prior to creating a password. In contrast, people do not usually have memorized bend sequences that they can use when creating a bend password. This describes the memorability difference between bend passwords and numeric passwords. On the other hand, bend passwords were faster to log in than PINs and had similar number of errors and perceived user satisfaction. Also, the deformation interaction was more accessible than touch-screen interaction, as it uses a more tactile input method and provides vibration as feedback. These learnings indicate the opportunities and limitations of bend passwords when comparing their usability with PIN passwords.

### 6.6 Bend Passwords in Future Devices

While the current prototype was a separate device, hence would require the user to carry an extra device, the paper does not endorse bend authentication to be implemented as such in future devices. Instead, we cater for expected near-future flexible phones as a vast amount of relevant work envisions that bend interaction will soon be integrated in mobile devices, thus we explore the authentication through bends accordingly. We envision that bend passwords would be integrated in such future flexible smartphones, or through a flexible phone case over a rigid smartphone (e.g. [21, 34]).

## 7 CONCLUSION

We explored the application of a bendable user authentication method for people with vision impairment using bend passwords. We conducted a user study with 16 people who are blind or have low vision to evaluate the learnability and memorability of bend passwords on BendyPass when compared to PINs on a smartphone. Our paper extends previous work, not only by engaging experts and participants with vision impairment, but also by challenging prior work on whether we should be designing such deformable authentication methods or not. We do not argue that adopting bend passwords is not necessarily the answer to touch-screen accessibility, but explore its potential and limitations as an alternative.

In this study, we gained insights from situating BendyPass in the hands of its intended users and learned about its ease of use, memorability, learnability and user satisfaction. We found that bend passwords do not outperform touch-based interactions for people with vision impairment. Despite being as easy to learn as PINs, bend passwords are still not easy to memorize. Bend passwords were significantly faster to enter than PINs on iPhone (using the screen reader VoiceOver and the standard typing style), but were not rated by users to be more secure than PINs. Such findings shed light on limitations of the bend interaction opportunity that was often over-promised in HCI literature.

Our inquiry was to assess how people with vision impairment will use bend passwords versus PIN passwords they already use. With accessibility applications, the key tenant is not outperformance, but providing people with multiple ways to achieve their goals in case one is ill-fitted. For example, voice commands are not faster than touch, but in some scenarios, or for some users, they are more

practical and it is useful to have the option to switch between interaction techniques that do not necessarily outperform each other altogether e.g. mouse and keyboard. Therefore, our paper is not claiming that Bend outperforms PIN or that having a clear winner is a goal, but is exploring the usability of each method to contribute to other researchers and designers what refinements can provide accessibility choices and create alternatives.

We envision deformability will be integrated in smartphones soon and argue that careful considerations should be taken into account and challenged when designing such interactivity. Future work should address challenges of deformable interaction for accessibility *in-use*, with the target user group, including association, memorability and security against shoulder surfing attacks, beyond the initial work on this subject [33]. We plan to explore further deformable gestures, replacing bend and fold by other deformations. We also plan to integrate it in the phone case, and connected to users' smartphones through Bluetooth. Such enhancements will allow a longitudinal study investigating the long-term usability of deformable authentication for people living with vision impairment, seeking the development of current hindering technology.

### ACKNOWLEDGMENTS

This work was supported and funded by the National Sciences and Engineering Research Council of Canada (NSERC) through a Discovery grant (2017-06300), a Discovery Accelerator Supplement (2017-507935) and the Collaborative Learning in Usability Experiences Create grant (2015-465639). We thank Kim Kilpatrick and Nolan Jenikov from the Canadian Council of the Blind in Ottawa for their great help.

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
