# OpenReview forum: "Bend or PIN: Studying Bend Password Authentication with People with Vision Impairment"
_graphicsinterface.org/Graphics_Interface/2020/Conference — GI 2020_

### Official Review · AnonReviewer1 · 2020-01-05
**Cool idea**

**Confidence:** 4
**Rating:** 7

**Review:**

This is an interesting and well-written paper which proposes a hardware device for entering tactile passwords ("Bendypass"). The target audience is blind or vision-impaired individuals that might find entering a traditional PIN or password too visible (and thus less secure) on their screens.

I liked the physical design of the device, and appreciated the short exploration into the device's design process. I found the Bendypass to be a clever and relatively simple idea built on top of this platform. Users found value in it, and it by and large performed about on par with traditional PINs.

One concern I had was with the analysis of the login time. The paper notes that "we selected the fasted login time", and yet the average (fastest) PIN time is 18.27s. This seems very high - if these were all 6-character passwords, that's an average of 3s per key. I would encourage the authors to double check this result, and if it holds, to explain if any outliers significantly affected the data.

It seems cumbersome to have to carry around a separate device just to unlock your phone. This could be made palatable if there were more ways to use the device, or if the functionality could be baked into the phone. Either way, the authors could expand on the practical considerations a bit more.

Overall, nifty idea, good prototype and good evaluation with the target population. I am therefore in favour of accepting this work.

---

### Official Review · AnonReviewer3 · 2020-01-09
**New knowledge about how users who are visually impaired can enter passwords using bendable interfaces**

**Confidence:** 5
**Rating:** 7

**Review:**

The submission presents evaluation of BendyPass, a prototype based on Bend Passwords design [33], with visually impaired people. The prototype is a simplified version of Bend Passwords [33] geared towards users who are visually impaired. The evaluation consisted of two sessions (taking place one week apart) in which participants first created their passwords and then used them to “sign in”. The experiment compared BendyPass with standard PIN security feature on  touchscreen devices.  The results show that although it took longer for participants to create their passwords with BendyPass, they were able to recall and enter them quicker with BendyPass than with PIN. This submission contributes new knowledge about how users who are visually impaired can enter passwords.

The main strength of the paper is the experimental user study design with users who are visually impaired. It is particularly important to evaluate technology with target stakeholders. The paper is well written: the work is motivated well, the related work is mostly comprehensive, and the design and evaluation sections are clear and have enough detail for others to attempt to reproduce/replicate the study.

However, there are two main weaknesses: 1) the submission narrowly focuses on bend passwords, and 2) the evaluation compares BendyPass against only one baseline.

The paper never justifies why Bend Passwords [33] is the best design to adapt for users who are visually impaired. There are many other potential designs out there and the paper does not fully explore the potential design space before picking Bend Passwords [33]. For example, an equally feasible alternative is a design that uses a small physical numerical keyboard that users can carry with them and enter passwords even from their pockets (the haptic feedback that such a keyboard would enable would allow such interaction). Such alternative design is similar to BendyPass along many dimensions (e.g., users need to carry an additional device, but offers a more familiar interface). Other designs exist (e.g., work by Das et al. (2017) is just one example.  Thus, the paper should better position  the proposed design/prototype within this design space.

This brings up another issue: the PIN baseline is the current de facto standard, but other baselines (e.g., physical PIN from the previous paragraph) would position the work better and help justify use of BendyPass’ very different and unfamiliar interaction modality. Also, entering PIN on  touchscreen devices is notoriously difficult for people who are visually impaired, so it is no wonder that BendyPass outperforms it. Thus, ideally the evaluation would compare other ways that participants can enter PIN passwords.

In summary, this is an interesting paper that will contribute to the GI community.  Thus, I look forward to seeing this paper as part of the program.

REFERENCES
Sauvik Das, Gierad Laput, Chris Harrison, and Jason I. Hong. 2017. Thumprint: Socially-Inclusive Local Group Authentication Through Shared Secret Knocks. In Proceedings of the 2017 CHI Conference on Human Factors in Computing Systems (CHI ’17). Association for Computing Machinery, New York, NY, USA, 3764–3774. DOI:https://doi.org/10.1145/3025453.3025991

---

### Official Review · AnonReviewer2 · 2020-01-11
**Interesting idea; study executed well but with concerns re: reporting and analysis; limited engagement with literature**

**Confidence:** 5
**Rating:** 5

**Review:**

The paper could use a more extensive engagement with prior research on non-visual authentication mechanisms for accessibility, e.g. Azenkot (cited, but not extensively discussed beyond simply a passing mention), or following surveys of similar literature (e.g. Helkala, K., 2012. Disabilities and authentication methods: Usability and security), or drawing from other research on alternative models of authentication (e.g. Aly, Y., 2016.  Spin-lock gesture authentication for mobile devices), or in general from work on security/authentication in the context of accessibility (e.g. work by Shirali-Shahreza on accessible CAPTCHA).

In particular I would have liked to see closely-related work such as that by Azenkot being engaged with more critically and more deeply, including in terms of anchoring the finding of the study in this (and other related) research. This is not present in the paper to the extend it could be to enhance the value of the contribution.

The motivation for the actual solution is very limited, and not grounded in prior work. This is not to say that this is not a clever design idea, but rather that the way the design is justified is not sufficient.

The evaluation study is rather limited, and does not compare this with other prior solutions for accessible authentication (for example, the PassChords system by Azenkot). To a minimum, I expect some very detailed / introspective analysis of the proposed system (and the results of the evaluation study) in comparison to such prior work. Similar to the research done by Azenkot, why were no other baselines considered? It seems that the baseline used had some accessibility features (the use of VoiceOver) but it's not clear how was this combined with the external pad? Why was a baseline fully contained within the phone that offered accessible authentication not considered? (Azenkot used the iPhone's built-in Passcode Lock with VoiceOver). I suggest that the authors clarify this part of the paper.

The participants' details are also lacking some crucial data. For example, how many are familiar with alternative authentication methods? How many use for example a gesture-based physical lock or other form of tactile lock in real life? What was their overall phone use, and mobile proficiency in general?

There are no details presented related any ethical considerations about this study. (and I consider that simply stating that the study was approved by an ethics office is not sufficient)

The interview data was analyzed through grounded theory -- more details are needed about how this was carried out (how many independent coders, how were the codes reconciled, how many codes, how were these grouped in themes, what was the resulting thematic map?) I am actually surprised that the authors deemed grounded theory to be necessary vs simply complementing their quantitative findings with quotes from participants.

The discussion should also include some consideration of the level of security afforded by the proposed method. While a full threat analysis model may be too much for a paper focused on usability, some discussion is still warranted -- after all, if the proposed method is significantly weaker security-wise, then there is not much point in proposing it as an alternative to current mechanisms.

While the statistical analysis seems correct, I would suggest some limitations are mentioned, as 18 participants is on the low end for such parametric tests to be meaningful. The reporting of statistical results does not include details provided about any corrections or any other distribution-based tests that may have been employed, and a justification for the selection of the analysis method. Finally, there are no hypotheses being provided, nor any justification for why this was the case.

---

### Meta-Review · Area_Chair1 · 2020-01-11

**Recommendation:** Accept
**Confidence:** 5

**Metareview:**

Reviewers agree that this paper has the potential to bring an interesting contribution, and that the research is mostly well executed. There are some concerns expressed by reviewers, mainly:
- Limited engagement with literature (R2) and with prior work in terms of authentication types (R2, R3)
- Limitations of the study re: baseline (R2, R3)
- Concerns re: analysis of data (R1, R2) and need to discuss limitations of analysis (R2, R3)
- Lack of discussion of ethical considerations (R2)
- Motivation for chosen design not well argued (R2, R3)

I believe some of these can be addressed with some editing

---

### Decision · Program_Chairs · 2020-01-11

Accept